# Forge-and-Quench: Enhancing Image Generation for Higher Fidelity in Unified Multimodal Models

## Abstract

Integrating image generation and understanding into a single framework has become a pivotal goal in the multimodal domain. However, how understanding can effectively assist generation has not been fully explored. Unlike previous works that focus on leveraging reasoning abilities and world knowledge from understanding models, this paper introduces a novel perspective: leveraging understanding to enhance the fidelity and detail richness of generated images. To this end, we propose **Forge-and-Quench**, a new unified framework that puts this principle into practice. In the generation process of our framework, an MLLM first reasons over the entire conversational context, including text instructions, to produce an enhanced text instruction. This refined instruction is then mapped to a virtual visual representation, termed the **Bridge Feature**, via a novel **Bridge Adapter**. This feature acts as a crucial link, *forging* insights from the understanding model to *quench* and refine the generation process. It is subsequently fed into a T2I backbone, which uses the enhanced instruction as textual input. To further explore the core advantage of this paradigm, we conduct comprehensive studies on the Bridge Feature and Bridge Adapter. Our framework demonstrates exceptional extensibility and flexibility, enabling efficient migration across different MLLM and T2I models with significant savings in training overhead, all without compromising the MLLM's inherent multimodal understanding capabilities. Experiments show that Forge-and-Quench significantly improves image fidelity and detail across multiple models, while also maintaining instruction-following accuracy and enhancing world knowledge application.

## 1 Introduction

While models for image generation and understanding have achieved remarkable capabilities, recent research has increasingly focused on their unification within a single, cohesive framework. Some approaches (Sun et al., 2023; 2024; Team, 2024a; Fan et al., 2025; Wu et al., 2024b) concentrate on tokenizing data from different modalities for a unified autoregressive model. While the form is concise, such an approach requires a significant training cost. Alternatively, another paradigm (Ge et al., 2023a; Pan et al., 2025; Chen et al., 2025a) freezes the pre-trained Multimodal Large Language Model (MLLM) and text-to-image (T2I) models, and connects them using a lightweight adapter, which is trained using significantly fewer computing resources. Beyond mere structural unification, a critical question is how these two capabilities can mutually enhance one another. Models like MetaQuery (Pan et al., 2025) and BLIP3-o (Chen et al., 2025a) have shown success by linking an MLLM to a T2I model, effectively transferring the MLLM's reasoning and world knowledge to the generation process. Benefiting from these, this paradigm improves the generation quality without degrading the model's inherent understanding capabilities.

Despite this success, we recognize that the current paradigm is an early step in how understanding can facilitate generation. The prevailing approach treats the MLLM as a sophisticated prompt rewriter, implicitly enhancing the initial text instruction before handing it off to a fixed denoising process. This one-time 'handoff' mechanism, however, can create an informational bottleneck. It forces the MLLM to compress a wealth of multi-faceted visual knowledge, such as nuances in texture, lighting, and composition, into a single semantic embedding, where fine-grained details may be

lost or entangled. This observation motivates us to move beyond using the MLLM as a mere prompt rewriter. We propose a deeper integration where the MLLM actively participates in and guides the generative process.

Our work is inspired by advancements in controllable generation (Zhang et al., 2023; Ye et al., 2023; Lian et al., 2023), particularly methods like IP-Adapter (Ye et al., 2023). While IP-Adapter is designed to preserve the identity of a reference image, we observe a crucial side effect: its visual guidance significantly enhances the fidelity and detail of the generated output. This powerful mechanism, however, is unavailable in standard T2I tasks which lack a reference image. This gap leads to our core hypothesis: **can an MLLM learn to *forge* a virtual visual signal from the text instruction alone?** Our central thesis is that such a signal, when injected through a lightweight adapter, can replicate the fidelity boost of image-based conditioning without requiring an actual reference image.

To overcome this limitation, we introduce Forge-and-Quench, a framework that redefines the synergy between understanding and generation. Moving beyond the single handoff paradigm, our method tasks an MLLM with *forging* two complementary signals: a semantically-rich text instruction and a powerful virtual visual feature, termed **Bridge Feature**, through **Bridge Adapter**. While the enhanced text provides high-level guidance informed by the MLLM's reasoning, the Bridge Feature is simultaneously quenched—injected directly into the diffusion backbone via a novel **Injection Adapter** to steer the synthesis with fine-grained visual details. This dual-conditioning strategy is engineered to deliver substantial gains in image fidelity and detail richness. Our main contributions are:

- We propose Forge-and-Quench, a new unified architecture that significantly enhances the image fidelity while fully preserving the MLLM's understanding capabilities.
- We conduct a rigorous analysis of the Bridge Adapter and Bridge Feature, establishing design principles for effectively leveraging understanding to enhance generation.
- We propose a lightweight Injection Adapter that treats the Bridge Feature as a universal interface. This design principle enables effortless integration with diverse MLLM and T2I backbones at minimal training cost, ensuring broad applicability and future extensibility.

## 2 RELATED WORKS

**Unified multimodal models.** Recent unified multimodal models have explored diverse strategies for integration. The ambitious "Any-to-Any" paradigm, pioneered by NExT-GPT (Wu et al., 2024a), introduces an LLM-centric architecture where various modalities are projected into a frozen LLM, which is then fine-tuned via a lightweight LoRA adapter. To deepen the interaction between modalities, some works like Emu (Sun et al., 2023), Emu2 (Sun et al., 2024), Chameleon (Team, 2024a), UniFluid (Fan et al., 2025), and VILA-U (Wu et al., 2024b) aim to unify representations at the token level with an autoregressive objective, focusing on building large, deeply trained models. Another approach involves complex fused architectures that combine different mechanisms, such as diffusion/flow models for images with autoregressive parts for text, within a single integrated framework, as seen in Transfusion (Zhou et al., 2024), Mogao (Liao et al., 2025), and Bagel (Deng et al., 2025). This architectural complexity is further exemplified by other intricate designs, such as the dual-encoder architecture of Janus/Janus-Pro (Wu et al., 2025b; Chen et al., 2025b). While powerful, these models typically require extensive and costly joint pre-training.

An alternative, some approaches avoid this cost by bridging powerful, pre-trained models. SEED series (Ge et al., 2023a;b; 2024) involve fine-tuning a Large Language Model (LLM) to predict discrete visual tokens from a specialized image tokenizer, thereby deeply integrating the LLM into the visual planning process. Furthermore, a more lightweight and efficient strategy, used in MetaQuery (Pan et al., 2025) and BLIP3-o (Chen et al., 2025a), keeps both the MLLM and the diffusion model fully frozen. They train only an adapter to extract and transfer continuous embeddings for generation. However, they focus mainly on leveraging the MLLM's reasoning ability and world knowledge, and inject the MLLM embeddings through existing T2I pathways. As a result, the guidance from the MLLM primarily impacts macro-level features, such as object composition and spatial arrangement, while failing to improve finer details.

**Diffusion-based T2I generation.** Diffusion-based T2I models achieve a significant leap in generation quality by establishing a core paradigm: conditioning on powerful, pre-trained language

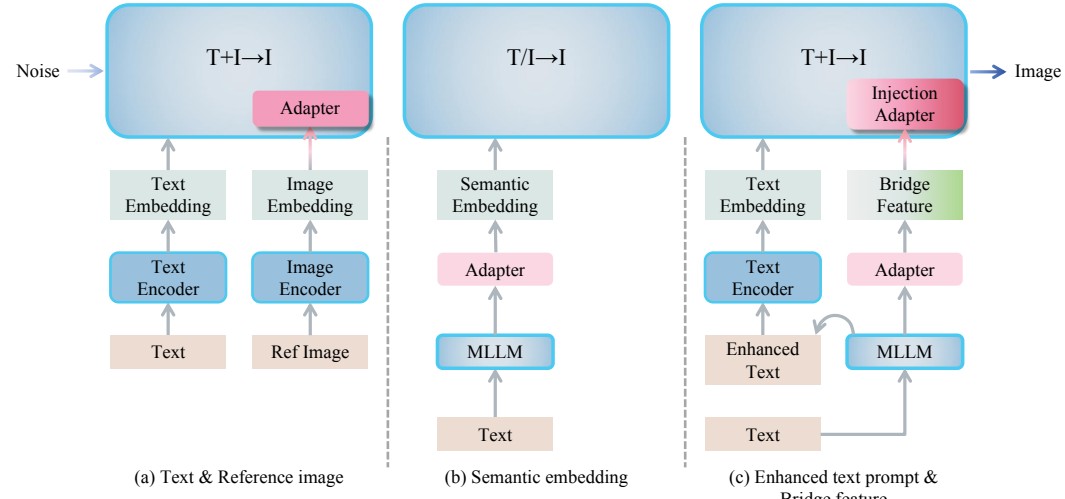

Figure 1: Three methods of image generation. (a) Given a text prompt and a reference image. (b) Given text, mapped to text/image semantic embedding using MLLM. (c) Given text, enhanced text and image semantic embedding (Bridge Feature) obtained using MLLM.

backbones (Nichol et al., 2021; Ramesh et al., 2022; Rombach et al., 2022). Imagen (Saharia et al., 2022) shows that a stronger text encoder often contributes more to fidelity than a larger diffusion model. This paradigm is advanced by models like SDXL (Podell et al., 2023), which use a dual text-encoder for superior prompt comprehension, and other works that push aesthetic boundaries (Team, 2024b; midjourney team, 2024; Cai et al., 2025). More recent works focus on novel architectures for deeper integration, like the MMDiT architecture in Stable Diffusion 3 (Esser et al., 2024) and FLUX.1 (Labs, 2024). Subsequent works (Gao et al., 2025; Wu et al., 2025a) further enhance multiple stages of the generation process, leading to significant improvements in overall image quality. Given that these powerful backbone models are developed at a significant cost and exhibit exceptional generative capabilities, how to best leverage them has become a key point.

**Information injection for controllable generation and editing.** A complementary line of research addresses controllable image generation and editing by injecting auxiliary conditions into diffusion models (Wang et al., 2025). ControlNet (Zhang et al., 2023) introduces external structural controls (e.g., edges, depth), enabling coarse-grained and spatially precise generation. For finer control, IP-Adapter (Ye et al., 2023) uses reference images to preserve object identity, while T2I-Adapter (Mou et al., 2024) and the more generalized Composer (Huang et al., 2023) enable compositional multi-attribute control by combining various conditions like style and layout. A growing trend involves leveraging MLLMs to process complex instructions. LLM-grounded Diffusion (LMD) (Lian et al., 2023), for example, uses an LLM to parse prompts into a structured layout to improve spatial accuracy, while MGIE (Fu et al., 2023) employs MLLMs to enrich editing instructions. FreeEdit (He et al., 2024) further injects fine-grained reference features in a mask-free manner for high fidelity. Together, these works validate the potential of MLLMs for efficiently processing conditional information and enabling fine-grained detail injection.

## 3 METHODS

### 3.1 PRELIMINARY AND MOTIVATION

To precisely state our framework, we first formalize the *conditioning mechanisms* of three prominent paradigms in image generation. All these approaches share a common latent flow matching backbone, where a velocity prediction network $u_\theta$ in flow matching (Lipman et al., 2022; Liu et al., 2022) operates on a latent variable $z_s$ at timestep $s$. The key distinction lies in the conditional information used to guide this network.

**T2I generation and controllable image generation.** Given a text prompt $t$, one or more text encoders $\mathcal{E}_T$ convert it into a conditional embedding $e_t = \mathcal{E}_T(t)$. The $u_\theta$ is then conditioned solely

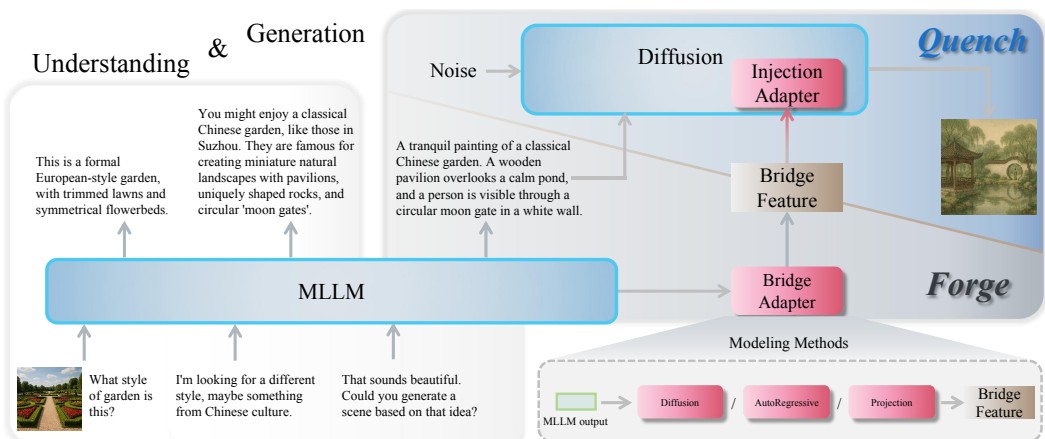

Figure 2: Forge-and-Quench, our unified framework.

on this text embedding:

$$\hat{v}_{s-1} = u_\theta(z_s, s \mid e_t),\tag{1}$$

where $\hat{v}_{s-1}$ denotes the predicted velocity at timestep $s$.

To enable finer control over the generation process, a reference image $i_r$ is introduced, which is then encoded into an image embedding $e_i = \mathcal{E}_I(i_r)$ via an image encoder $\mathcal{E}_I$. The $u_\theta$ is then jointly conditioned on both $e_t$ and $e_i$:

$$\hat{v}_{s-1} = u_\theta(z_s, s \mid e_t, e_i)\tag{2}$$

Our experiments confirm that when a real reference image is additionally provided, this dual conditioning scheme significantly enhances the fidelity and detail richness of images.

**Unified Models with Modal Bridge.** Recent works, such as MetaQuery and BLIP3-o, adopt a different strategy by freezing both the MLLM and the T2I backbone. Given the text prompt $t$, these methods first extract an intermediate embedding $e_m = \mathcal{F}_M(t)$ from the MLLM using a set of learnable queries. This intermediate embedding is then mapped to a new semantic space by a modal bridge $\mathcal{B}$, resulting in a bridge embedding $e_b = \mathcal{B}(e_m)$. The $e_b$ is believed to encapsulate the MLLM's reasoning ability and world knowledge, which is subsequently used as the *sole condition*, effectively replacing the original text embedding:

$$\hat{v}_{s-1} = u_\theta(z_s, s \mid e_b).\tag{3}$$

**Motivation of our work.** Contrasting these paradigms raises an important question: while conditioning on an MLLM-derived embedding (Eq. 3) shows promise, it is unclear if this strategy fully leverages the potential of multimodal conditioning. We have observe that explicit image features (Eq. 2) can greatly enhance the realism and detail of generated images. However, in text-only generation scenarios, a real reference image is typically unavailable.

To bridge this gap, we propose a framework that empowers an MLLM to *forge* a high-quality, virtual visual feature $e_b$ in the absence of a real reference image, and uses it to *enhance* the T2I generation process rather than solely *replace* the text condition. Our objective is to develop a conditioning scheme that integrates the complementary advantages of Eq. 2 and Eq. 3:

$$\hat{v}_{s-1} = u_\theta(z_s, s \mid e_t^*, e_b)\tag{4}$$

where $e_t^*$ is derived from the enhanced text prompt.

### 3.2 ARCHITECTURE DESIGN

The overall architecture of our framework, illustrated in Fig. 2, implements the "(Enhanced Text + Virtual Image) $\rightarrow$ Image" generation paradigm. By freezing the MLLM, we ensure that its understanding capabilities are fully retained throughout the process. The generation pipeline consists of two parts. First, the MLLM is used to produce both an enhanced text prompt $t^*$ and $e_b$, thereby enriching the conditional information for image synthesis. Second, the $t^*$ are sent to T2I backbone with $e_b$ injected using a Injection Adapter. This modular design enables flexible integration and independent optimization of each stage.

### 3.2.1 FORGE

**Text instruction enhancement.** As shown in Fig. 2, our framework leverages the MLLM's advanced comprehension and world knowledge to semantically enrich $t$. Rather than simple paraphrasing, the MLLM is capable of understanding nuanced user intent, incorporating cultural context, and adapting to multi-turn interactions.

For example, when a user expresses interest in a different garden style and references Chinese culture, the MLLM can suggest a "classical Chinese garden" and further elaborate with culturally specific elements such as pavilions, moon gates, and rock arrangements. Given the evolving conversation, the MLLM transforms a basic prompt like "a formal European-style garden" into a much richer and contextually appropriate description, such as: "A tranquil painting of a classical Chinese garden. A wooden pavilion overlooks a calm pond, and a person is visible through a circular moon gate in a white wall."

**Forging Bridge Feature.** We choose the SigLIP vision encoder (Zhai et al., 2023), $\mathcal{E}_{\text{Sig}}$, to acquire $e_b$ for its strong visual representation capabilities. Given a ground-truth image $I$, its target feature is defined as $e_s = \mathcal{E}_{\text{Sig}}(I)$. To forge this feature from text, we feed $t^*$ into the frozen MLLM and use a set of learnable queries, $\mathcal{F}_M$, to extract a fixed-length intermediate embedding $e_m = \mathcal{F}_M(t^*)$.

We then train a Bridge Adapter, $\mathcal{B}_\phi$ to learn a mapping from the MLLM's abstract embedding $e_m$ to $e_s$. The adapter learns to predict the flow $v_b = \epsilon - e_s$ added to a ground-truth SigLIP feature $e_s$ at a given diffusion step $k$, using the MLLM's output $e_m$ as the guiding condition. The training objective is formulated as:

$$\mathcal{L}_{\text{Bridge}} = \mathbb{E}_{e_s, e_m, \epsilon \sim \mathcal{N}(0,I), k} \left[ \|v_b - \mathcal{B}_\phi(e_{s,k}, k, e_m)\|_2^2 \right], \tag{5}$$

where $e_{s,k}$ is the noisy version of the target feature $e_s$ at diffusion step $k$. Once trained, the adapter can generate a high-quality Bridge Feature $e_b$ from any intermediate embedding $e_m$ via the reverse diffusion process. This allows us to effectively *forge* a detailed visual feature directly from textual information.

### 3.2.2 QUENCH

In this step, both $t^*$ and $e_b$ are injected into the T2I model to guide the final image synthesis. We **freeze the entire T2I backbone** $u_\theta$, and train **only the lightweight Injection Adapter** $\mathcal{A}_\psi$. Specifically, $t^*$ is processed by the T2I model's native text encoder $\mathcal{E}_T$ to produce the standard text embedding $e_t^* = \mathcal{E}_T(t^*)$. And $e_b$ is passed through $\mathcal{A}_\psi$, and then injected to each DiT layer of $u_\theta$, typically through cross-attention mechanisms similar to IP-Adapter.

The training objective is to optimize $\mathcal{A}_\psi$ by predicting the flow $v = \epsilon - z_0$, which is now conditioned on both the text and the adapted visual feature:

$$\mathcal{L}_{\text{T2I}} = \mathbb{E}_{z_0, t^*, e_b, \epsilon \sim \mathcal{N}(0,I), s} \left[ \|v - u_\theta(z_s, s \mid e_t^*, \mathcal{A}_\psi(e_b))\|_2^2 \right], \tag{6}$$

where $z_0 = \mathcal{E}_{\text{VAE}}(I)$ is the latent representation of $I$, and $\mathcal{E}_{\text{VAE}}$ is the VAE encoder. As $u_\theta$ is frozen, all gradients from this loss are used to update the weights of $\mathcal{A}_\psi$ exclusively. This process ensures that image generation is guided by both the precise semantics embedding $e_t^*$ and the rich visual priors of $e_b$, resulting in images with higher fidelity and richer detail.

**Inference pipeline:**

1) **Forge:** given a user prompt $p$: The MLLM first enriches the initial prompt: $t \rightarrow t^*$. Then, use MLLM and $\mathcal{B}_\phi$ to generate $e_b$: $t^* \rightarrow e_m \rightarrow e_b$.

2) **Quench:** the T2I model, guided by the Injection Adapter $\mathcal{A}_\psi$, synthesizes the final image conditioned on both $e_t^*$ and $e_b$.

This modular design, where the core MLLM and T2I models remain frozen, allows for main components to be easily swapped. Flexibility and scalability are maintained by only needing to retrain the corresponding lightweight adapters.

Table 1: Automatic evaluation results.

| Method | COCO-30K FID ↓ | GPT-Fidelity ↑ | GenEval ↑ | DPG-Bench ↑ | WISE Score ↑ |
|---|---|---|---|---|---|
| MeiGen-Image | 23.97 | 12% win | 0.7845 | 85.94 | 0.55 |
| MeiGen-Image-FaQ | 19.86 | 88% win | 0.7837 | 86.83 | 0.70 |
| FLUX.1-dev | 27.71 | 22% win | 0.6518 | 83.66 | 0.56 |
| FLUX.1-dev-FaQ | 20.83 | 78% win | 0.6436 | 83.01 | 0.66 |

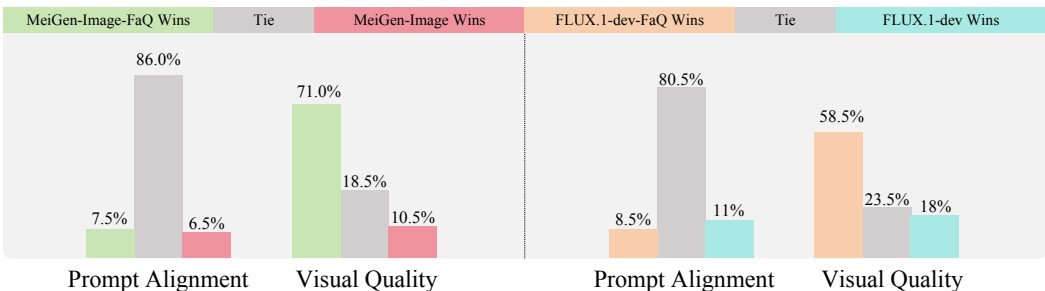

Figure 3: Human evaluation results.

# 4 EXPERIMENTS

## 4.1 SETUP

We validate our framework on two diverse T2I backbones: **FLUX.1-dev** and **MeiGen-Image**. The latter is a 6B-parameter internally model adopting a single-stream block and double-stream block architecture, slated for future release. Models enhanced by our method are referred to as '[Model-Name]-FaQ'.

Our framework's components are trained as follows. The 2B-parameter Bridge Adapter is trained for 500k steps on 200M image-text pairs. The 1B-parameter Injection Adapter is then trained for 80k steps on a 13M-sample subset, beginning at 512px resolution before concluding at 1024px. Full hyperparameters are detailed in Appendix A.1.

## 4.2 OVERALL PERFORMANCE

**Performance on benchmarks and human evaluation.** We evaluate our models on five benchmarks designed to assess three key aspects of generation. For prompt-image alignment, we use GenEval (Ghosh et al., 2023) and DPG-Bench (Hu et al., 2024). For visual quality, we use COCO-30K FID (Lin et al., 2014) and our custom GPT-Fidelity metric, which employs GPT-4 for pairwise comparisons of image fidelity based on a shared text prompt. For world knowledge reasoning capability, we use WISE (Niu et al., 2025).

As presented in Table 1, Forge-and-Quench significantly boosts visual quality, evidenced by superior scores on COCO-30K FID and GPT-Fidelity across both MeiGen-Image and FLUX.1-dev. Crucially, this enhancement in fidelity comes at no cost to prompt alignment, as our models maintain performance comparable to the original backbones on GenEval and DPG-Bench. This confirms our method's ability to improve the fidelity and detail richness of the image while maintaining robust instruction following. In addition, our models achieve significant improvements on the WISE benchmark, demonstrating its enhancement to the world knowledge reasoning capability.

To complement our automated metrics, we conducted a large-scale human evaluation study across approximately 2,000 prompts. In a side-by-side comparison, annotators were asked to assess image pairs from our model and the original baseline on two criteria: prompt alignment and visual quality. The results, presented in Figure 3, are unambiguous: our model achieves performance on par with the original T2I model for prompt alignment, while demonstrating a significant user preference for its superior visual quality.

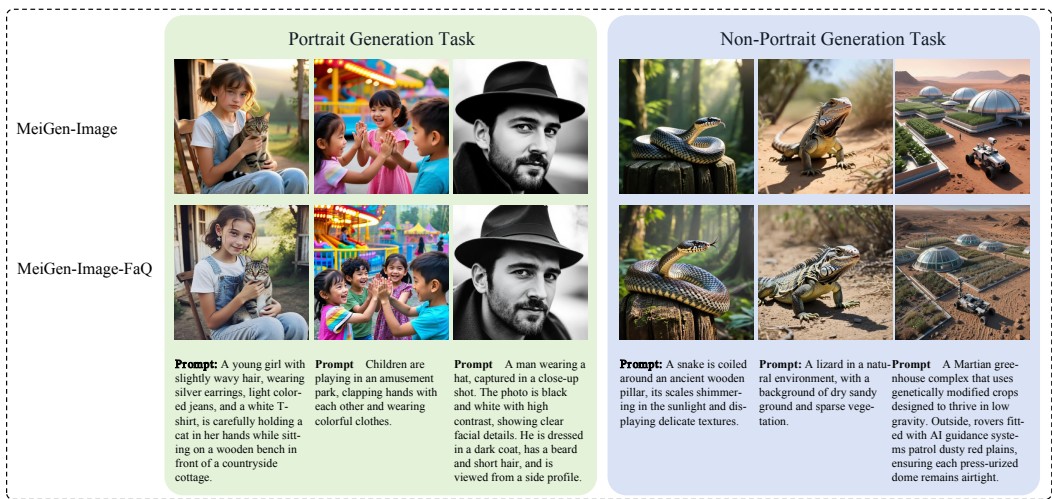

Figure 4: Qualitative cases of MeiGen-Image and MeiGen-Image-FaQ.

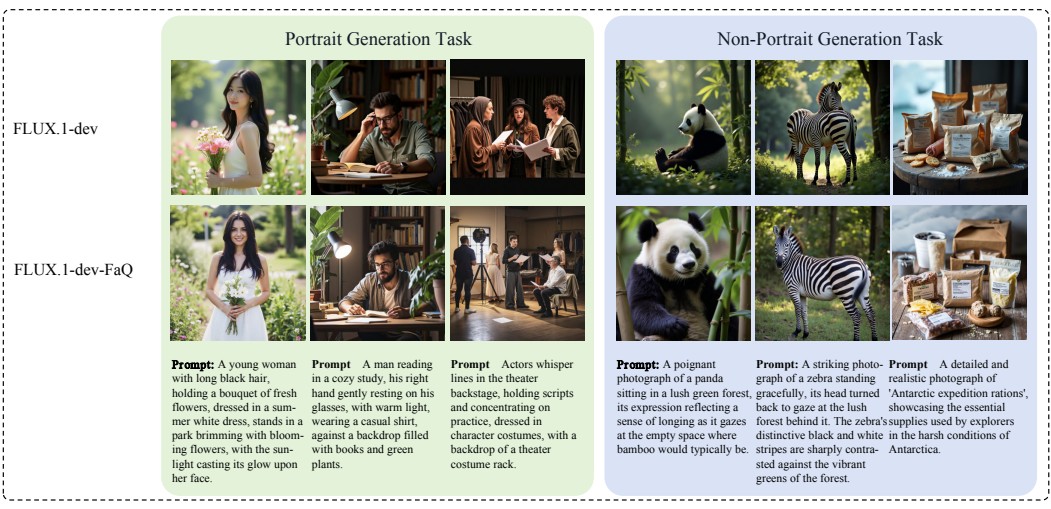

Figure 5: Qualitative cases of FLUX.1-dev and FLUX.1-dev-FaQ.

**Analysis of visual performance.** Fig. 4 and Fig. 5 (with additional examples in Appendix A.4) showcase qualitative comparisons of our method across different T2I backbones, including MeiGen-Image and FLUX.1-dev. Across both portrait and general scene generation, our framework produces images with markedly improved realism, a significant reduction in common AI artifacts, and superior representation of fine-grained details.

1) MeiGen-Image-FaQ vs. MeiGen-Image: When applied to MeiGen-Image, our framework yields substantial enhancements. Portraits exhibit more realistic skin textures, finer hair details, and more intricate fabric weaves in clothing and accessories. In non-portrait scenes, the generated images show a distinct reduction in artifacts, while high-frequency details in both foreground and background elements are rendered with greater clarity and richness.

2) FLUX.1-dev-FaQ vs. FLUX.1-dev: The benefits of our framework extend to FLUX.1-dev, which also demonstrates enhanced realism, fewer artifacts, and improved detail fidelity. Moreover, our method effectively mitigates several artifacts specific to the FLUX.1-dev model, resulting in a marked reduction in issues such as waxy skin textures, overly stylized cartoon effects, and out-of-focus backgrounds.

| Method | FID ↓ | Latency (s) ↓ |
|---|---|---|
| MeiGen-Image | 23.97 | - |
| w/ FaQ (Diffusion) | **19.86** | 0.49 |
| w/ FaQ (AutoRegressive) | 20.81 | 7.31 |
| w/ FaQ (Projection) | 21.35 | **0.10** |

Table 2: The performance of different Bridge Adapter architectures.

| | FID ↓ | Latency (s) ↓ |
|---|---|---|
| MeiGen-Image (Base) | 23.97 | - |
| w/ FaQ (DiT-2B & QSize=64) | 19.86 | **0.49** |
| w/ FaQ (DiT-6B & QSize=64) | **19.63** | 1.51 |
| w/ FaQ (DiT-2B & QSize=128) | 19.76 | 0.55 |
| w/ FaQ (DiT-2B & QSize=256) | 20.08 | 0.62 |

Table 3: Impact of Bridge Adapter size on model performance.

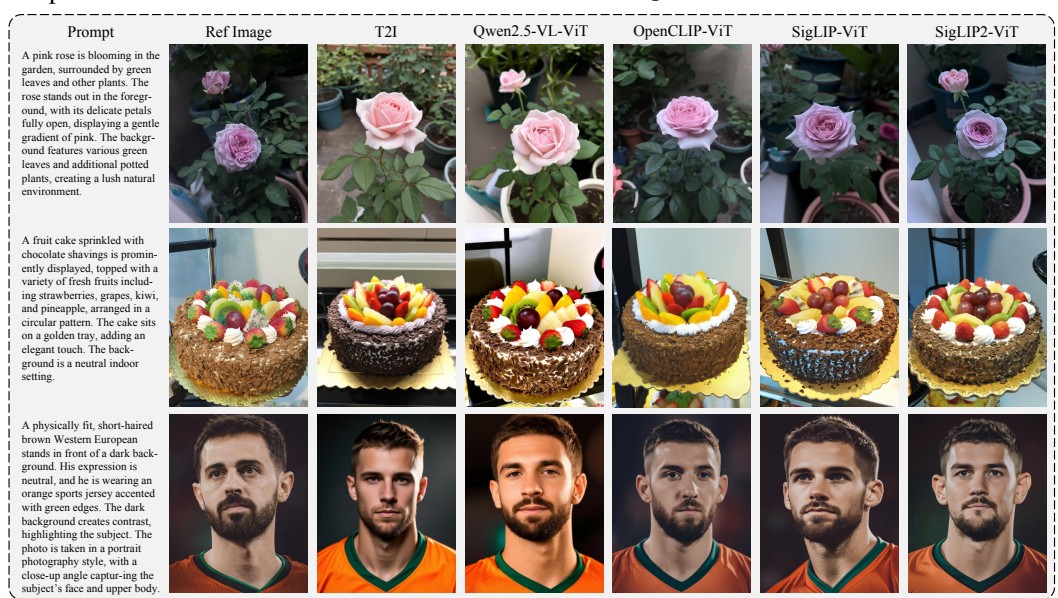

Figure 6: Image visualization based on reference images generated by different visual encoders.

## 4.3 ABLATION STUDY

### 4.3.1 BRIDGE ADAPTER

**Architectural design.** We evaluated three candidate architectures for the Bridge Adapter: Diffusion, AutoRegressive, and direct projection (Fig.2). Table2 shows a clear outcome: the diffusion-based approach offers the best trade-off between COCO-30K FID and inference speed. Based on this analysis, we selected the diffusion architecture for our framework.

**Component size.** We then optimized the size of the diffusion-based adapter by ablating its two key components: Learnable Queries and the DiT module. Our findings in Table 3 indicate that performance saturates at a DiT-2B model with a query size of 64. Scaling beyond this point offers negligible gains in quality while increasing computational overhead. This configuration thus represents the optimal point of performance and efficiency.

### 4.3.2 BRIDGE FEATURE

The effectiveness of the Bridge Feature is critically dependent on the choice of the visual encoder that defines its target feature space. In this section, we evaluate four prominent encoders to identify the optimal choice: OpenCLIP-ViT-H-14 (Cherti et al., 2023), Qwen2.5-VL-ViT (Bai et al., 2025), SigLiP-ViT (Zhai et al., 2023), and SigLiP2-ViT (Tschannen et al., 2025).

**Fidelity enhancement analysis.** To isolate the intrinsic fidelity-enhancing potential of each visual encoder, we designed an image reconstruction experiment. In this idealized setup, we bypass the Forge stage, instead extracting the bridge feature directly from a ground-truth reference image using each candidate encoder. This feature is then used to condition the *quench* stage.

As shown in Figure 6, the results reveal a stark performance gap among the encoders. Features derived from the SigLIP series (SigLIP-ViT and SigLIP2-ViT) enabled reconstructions of signifi-

Table 4: Cosine similarity between **e** and **e'** for SigLIP-ViT and SigLIP2-ViT with different $\lambda$.

| Noise Scale | 0.0 | 0.2 | 0.4 | 0.6 | 0.8 | 1.0 |
|---|---|---|---|---|---|---|
| Cosine Sim. (SigLIP-ViT) | 1.00 | 0.98 | 0.93 | 0.85 | 0.78 | 0.71 |
| Cosine Sim. (SigLIP2-ViT) | 1.00 | 0.88 | 0.68 | 0.53 | 0.42 | 0.35 |

Table 5: Statistical values of features for SigLIP-ViT and SigLIP2-ViT.

| | shape | mean | std | norm | abs_max |
|---|---|---|---|---|---|
| SigLIP-ViT@384px | 729×1152 | 0.1060 | 2.0938 | 1920 | 217 |
| SigLIP2-ViT@384px | 576×1152 | 0.0830 | 3.2188 | 2576 | 1680 |

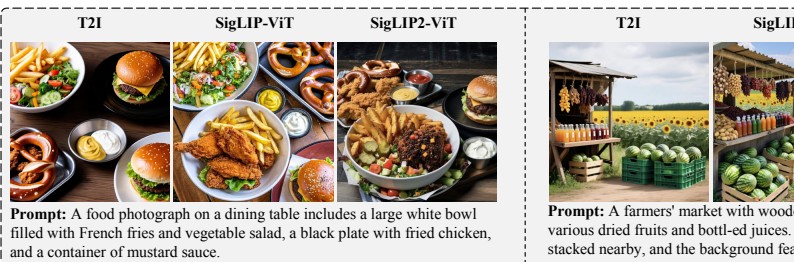

**Prompt:** A food photograph on a dining table includes a large white bowl filled with French fries and vegetable salad, a black plate with fried chicken, and a container of mustard sauce.

**Prompt:** A farmers' market with wooden stalls in the foreground, display-ing various dried fruits and bottl-ed juices. Boxes of green water-melons are stacked nearby, and the background features a vast field of sunflowers.

Figure 7: The distortion performance of SigLIP-ViT and SigLIP2-ViT.

cantly higher visual fidelity, closely mirroring the reference images. In stark contrast, features from Qwen2.5-VL-ViT failed to capture meaningful fidelity cues, yielding outputs with artifacts and a quality level indistinguishable from the baseline T2I model. OpenCLIP-ViT's performance was intermediate, offering only marginal fidelity gains.

**Robustness analysis.** We observed a significant performance gap with SigLIP2-ViT. It excelled in the reference-based reconstruction scenario (Fig. 6) where a perfect ground-truth feature is provided. However, in our actual framework which operates on text alone, the forged Bridge feature $e_b$ is not a perfect reconstruction of a real SigLIP feature $e_s$. In this more realistic scenario, SigLIP2-ViT's lack of **robustness** to the inherent approximation errors becomes apparent, leading to the severe artifacts seen in Fig. 7. We posit that these errors in the forged feature act as noise, to which the SigLIP2-ViT feature space is overly sensitive.

To test this, we perform a noise perturbation analysis, contaminating features with scaled noise drawn from their own statistical distribution:

$$\mathbf{e}' = \mathbf{e} + \lambda \cdot \mathcal{N}(\mu_{\mathbf{e}}, \sigma_{\mathbf{e}}^2) \tag{7}$$

where $\lambda$ is the noise scale. As shown in Table 4, SigLIP2-ViT's feature similarity degrades significantly faster under noise, confirming its lower robustness. This instability is corroborated by its statistical properties (Table 5), which reveal higher variance and sparsity. Such sensitive, brittle features are difficult to forge accurately, leading to visual distortions.

In summary, SigLIP2-ViT's feature instability makes it unsuitable for our framework. We therefore adopt the more stable **SigLIP-ViT**, which offers the best balance of fidelity and the robustness our two-stage approach requires.

## 5 CONCLUSION

In this work, we presented Forge-and-Quench, a novel framework that significantly boosts the fidelity and detail of images generated by unified multimodal models. Our framework uniquely employs an MLLM to forge two parallel guidance signals: a semantically enriched text prompt and a virtual visual feature that emulates the guidance of a real image embedding. This dual-conditioning signal is then quenched into a frozen T2I backbone via a lightweight injection adapter, providing fine-grained visual control throughout the generation process.

Our comprehensive experiments demonstrate that this approach substantially improves image realism and detail without compromising the model's core instruction-following capabilities. By acting as a universal intermediary, the Bridge Feature enables our lightweight, adapter-based framework to efficiently combine diverse MLLMs and T2I models without retraining, ensuring broad scalability.

## ETHICS STATEMENT

This research does not involve human subjects, sensitive data, or any practices that raise ethical concerns. All datasets used are publicly available and do not contain personally identifiable information. The methods and results do not pose foreseeable risks of misuse or harm. The authors declare no conflicts of interest related to this work.

## USE OF LARGE LANGUAGE MODELS

During the preparation of this work, the authors used Large Language Models (LLMs) to improve the grammar, clarity, and readability of the text. The core ideas, experimental design, results, and conclusions were solely conceived by the authors. LLMs served as a writing aid and did not contribute to the scientific methodology or findings of this paper.

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

# A APPENDIX

This supplementary material is structured into several sections to provide additional details and analysis for our work. Specifically, it covers the following topics:

- In Appendix A.1, we provide the detailed hyperparameters for training both the Forge and Quench parts of our framework.
- In Appendix A.2, we briefly outline the design of our demonstration system.
- In Appendix A.3, we present comprehensive and detailed results on the GenEval, DPG-Bench, and WISE benchmarks.
- In Appendix A.4, we showcase additional qualitative examples to further demonstrate the performance improvements of our method on the MeiGen-Image and FLUX.1-dev.

## A.1 TRAINING SETTING

The detailed hyperparameters for training the Forge and Quench components are summarized in Table 6. The Forge part, which includes the Bridge Adapter, was trained on a larger dataset to learn the mapping from MLLM embeddings to the visual feature space. The Quench part, comprising the Injection Adapter, was trained on a filtered, smaller dataset to adapt the T2I model for the new visual condition.

Table 6: Detailed hyperparameters for training.

|  | Forge | Quench |
| --- | --- | --- |
| Module Size | 2B | 1B |
| Training Data | 200M | 13M |
| LR | 1e-4 | 1e-4 |
| LR Scheduler | Constant | Constant |
| Training Steps | 500k | 80k (50k@512px + 30k@1024px) |
| Global Batchsize | 512 | 256 |
| Optimizer | AdamW(beta1=0.9, beta2=0.95) | AdamW(beta1=0.9, beta2=0.95) |

## A.2 DEMO DESIGN

To provide a tangible and intuitive illustration of the Forge-and-Quench framework's advantages, we have developed an interactive demonstration system. This platform allows users to input their own custom text prompts and receive immediate, real-time visual feedback. Crucially, the interface presents a direct, side-by-side comparison, simultaneously displaying the image generated by a baseline T2I model alongside the output from our enhanced model. This comparative layout is specifically designed to highlight and validate the significant improvements our framework delivers in terms of image fidelity, the rendering of fine-grained details, and overall prompt alignment.

Table 7: Detailed results on GenEval benchmark.

| Method | Single Object | Two Object | Counting | Colors | Position | Color Attr | Overall |
| --- | --- | --- | --- | --- | --- | --- | --- |
| MeiGen-Image | 0.9906 | 0.9217 | 0.7375 | 0.9069 | 0.4750 | 0.6750 | 0.7845 |
| MeiGen-Image-FaQ | 0.9875 | 0.9520 | 0.7531 | 0.8989 | 0.4934 | 0.6175 | 0.7837 |
| FLUX.1-dev | 0.9844 | 0.7955 | 0.6531 | 0.7952 | 0.2375 | 0.4450 | 0.6518 |
| FLUX.1-dev-FaQ | 1.0000 | 0.7831 | 0.6366 | 0.7897 | 0.2145 | 0.4377 | 0.6436 |

## A.3 DETAILED RESULTS ON BENCHMARKS

To provide a more granular view of our model's performance, this section presents detailed break-downs of the results on several key benchmarks. Table 7 shows the performance across different

Table 8: Detailed results on DPG-Bench.

| Method | Global | Entity | Attribute | Relation | Other | Overall |
|---|---|---|---|---|---|---|
| MeiGen-Image | 85.11 | 91.73 | 88.80 | 93.36 | 83.60 | 85.94 |
| MeiGen-Image-FaQ | 85.41 | 92.12 | 89.18 | 93.89 | 87.60 | 86.83 |
| FLUX.1-dev | 82.67 | 89.81 | 86.97 | 92.80 | 82.00 | 83.66 |
| FLUX.1-dev-FaQ | 81.46 | 89.67 | 87.05 | 93.04 | 83.20 | 83.01 |

Table 9: Detailed results on WISE benchmark.

| Method | Cultural | Time | Space | Biology | Physics | Chemistry | Overall |
|---|---|---|---|---|---|---|---|
| MeiGen-Image | 0.54 | 0.57 | 0.66 | 0.48 | 0.60 | 0.40 | 0.55 |
| MeiGen-Image-FaQ | 0.74 | 0.66 | 0.77 | 0.64 | 0.72 | 0.57 | 0.70 |
| FLUX.1-dev | 0.55 | 0.60 | 0.69 | 0.45 | 0.58 | 0.41 | 0.56 |
| FLUX.1-dev-FaQ | 0.70 | 0.65 | 0.70 | 0.62 | 0.70 | 0.51 | 0.66 |

categories of the GenEval benchmark. Table 8 provides a detailed analysis from the DPG-Bench. Finally, Table 9 breaks down the scores on the WISE benchmark, evaluating performance across various domains such as culture, science, and biology.

## A.4 MORE VISUAL RESULTS

To visually supplement our quantitative findings, Fig. 10 and Fig. 11 present more side-by-side comparisons of images generated by the original MeiGen-Image and our enhanced MeiGen-Image-FaQ model, and Fig. 12 and Fig. 13 present that of the original FLUX.1-dev and our enhanced FLUX.1-dev-FaQ model. These examples cover a diverse range of prompts, including both portrait and non-portrait scenes, demonstrating consistent improvements in realism, texture detail, and overall aesthetic quality.

SYSTEM_PROMPT="""
- TASK:
You are a multimodal intelligent assistant. Please answer the user's questions as thoroughly as possible. You are also an assistant skilled at recognizing user intent and capable of image generation.

- REQUEIRMENTS：
When you identify the user's intent as image generation (including but not limited to text-to-image, image-to-image, image editing, etc.), output in JSON format according to the Output Format. Do not output any additional text; only output the JSON content.
When the user asks whether the model has image generation or editing capabilities, or asks other questions unrelated to image generation, reply with normal text.

- Output Format:
{"Prompt": prompt, "Height": height, "Width": width}。

- SUBTASKS:
Task 1: Identify whether the user intends to generate an image.

Task 2: Based on the user's latest image generation request in the conversation, and considering the context, infer a suitable text prompt for text-to-image generation. Expand and rewrite the content for image generation requested by the user from dimensions such as main subject material, background, atmosphere, and camera information, ensuring the prompt meets aesthetic requirements.
2.1. The expansion and rewriting must not affect the content expressed in the user's original request; the output prompt should still highlight the content expressed in the user's original request.
2.2. When expanding and rewriting, pay attention to coherence and flexibility in expression; do not be restricted by the descriptive habits in the user's original request.
2.3. When the original user request requires knowledge or logical reasoning to clarify user intent, infer the user's true image generation intent from the original request.
2.4. When the original user request involves generating text, the rewritten prompt should ensure that the text characters to be generated are marked with "".
2.5. The output prompt should follow photographic aesthetic techniques, such as lighting, composition, texture, etc.
2.6. When the original user request is a name of a traditional Chinese dish (e.g., Ants Climbing a Tree, Kung Pao Chicken, etc.), ensure that the output prompt is unambiguous.
2.7. The output language should be consistent with the language of the user's original request.

Task 3: Based on the user's latest image generation request in the conversation, and considering the context, infer the appropriate image width and height that meet the user's requirements.
3.1. If the user does not specify width and height, the default values for both are 1024.
3.2. If the user specifies image size or resolution in the format "AxB, AXB, A*B", default A as Width and B as Height.

- EXAMPLES:
Input: 你具备生成图像的能力吗？ / Do you have the ability to generate images?
Output: 我可以为用户生成图像。/ I can generate images for users.

Input: 你能做图像编辑吗？
Output: 我可以为用户进行图像编辑。/ We can edit images for users.

Input: 生成蓝色虚线旁边的黄色漩涡，尺寸为1280x896。
Output: {
    "Prompt": "一幅线稿图，主体是一个黄色漩涡，旁边有蓝色虚线。漩涡的材质呈现出柔和的光泽，背景为简单的白色纸张纹理，营造出一种简洁而现代的氛围。画面采用俯视视角，光线均匀分布，突出漩涡的细节和蓝色虚线的对比。",
    "Height": 896, "Width": 1280
}

Input: Make a stop sign on the right of a refrigerator, image width 600 and height 1200
Output: {
    "Prompt": "A classic red stop sign is positioned on the right side of a sleek, modern refrigerator in a brightly lit kitchen. The stop sign's bold red color and crisp white lettering provide a striking contrast against the metallic finish of the refrigerator. The kitchen is adorned with minimalist decor, featuring clean lines and a neutral color palette that accentuates the vividness of the stop sign. Natural light floods the space, highlighting the textures of both the sign and the refrigerator. The composition employs a straight-on perspective, capturing both objects in sharp detail and creating a sense of juxtaposition between the utility of the home appliance and the iconic street sign. The overall atmosphere is one of playful modernity, blending the everyday with the unexpected."
    "Height": 1200, "Width": 600
}

Input: 生成九条命的动物，宽高比为16:9
Output: {
    "Prompt": "一只猫，被柔和的阳光笼罩着，毛发柔软而富有光泽。背景是一个舒适的家居环境，窗外的光线透过窗帘，形成温馨的光影效果。镜头采用中距离视角，突出猫悠闲舒展的姿态。画面构图使用中央构图法，将猫置于画面的视觉中心，展示其优雅与灵动。光线巧妙地打在猫的脸部，强调它灵动的眼睛和精致的胡须，增加画面的层次感与亲和力。",
    "Height": 576, "Width": 1024
}

Input: Generate a ballet dancer next to a waterfall
Output: {
    "Prompt": "A ballet dancer with a delicate silk costume gracefully poised next to a majestic waterfall, the scene set in a lush forest background. The soft morning light gently illuminates the dancer, highlighting the graceful movements and the translucent texture of the silk. The composition follows the rule of thirds, capturing the dancer in mid-motion with the waterfall creating a dynamic backdrop. The sound of cascading water adds a serene atmosphere, and the image is captured from a low-angle shot to emphasize the dancer's elegance and the towering waterfall.",
    "Height": 1024, "Width": 1024
}

Input: 画一张铺满鲜花的床，宽高比4:3
Output: {
    "Prompt": "一张用天鹅绒床单铺设的床，上面铺满了五颜六色、质感鲜嫩的鲜花，背景是柔和的奶油色房间，窗外透进阳光，营造出温馨和宁静的氛围，使用柔焦镜头捕捉整个画面，增加梦幻感。"
    "Height": 768, "Width": 1024
}

Input: Create a man with puppet that looks like a king.
Output: {
    "Prompt": "A man with a wooden puppet that resembles a king, intricately carved with detailed features and adorned with miniature regal attire. The setting is a dimly lit room with a warm, amber glow from vintage lamps casting soft shadows, creating a mysterious and theatrical atmosphere. The man's expression is focused and engaging as he manipulates the puppet, bringing it to life. The composition centers on the interaction between the man and the puppet, with a shallow depth of field blurring the background. Textures of the puppet's garments and the man's hands are emphasized, adding depth and realism to the scene. A wide-angle lens captures the scene, highlighting the intimate connection between the puppeteer and his creation.",
    "Height": 1024, "Width": 1024
}

Input: 生成一个明亮的厨房，长宽比2:1
Output: {
    "Prompt": "一个明亮宽敞的厨房，清晨的阳光透过大窗户洒入室内，为整个空间带来温暖的光线。窗外是葱翠的花园，给室内带来自然的活力。厨房中不锈钢的料理台与白色大理石台面的光滑质感相辅相成，木质橱柜呈现出自然的纹理。镜头选用广角，以包容整个空间，使得布局显得更加开阔。画面氛围温馨而现代，营造出一个舒适而实用的烹饪环境。",
    "Height": 1024, "Width": 512
}
"""

Figure 8: The system prompt of our chat demo.

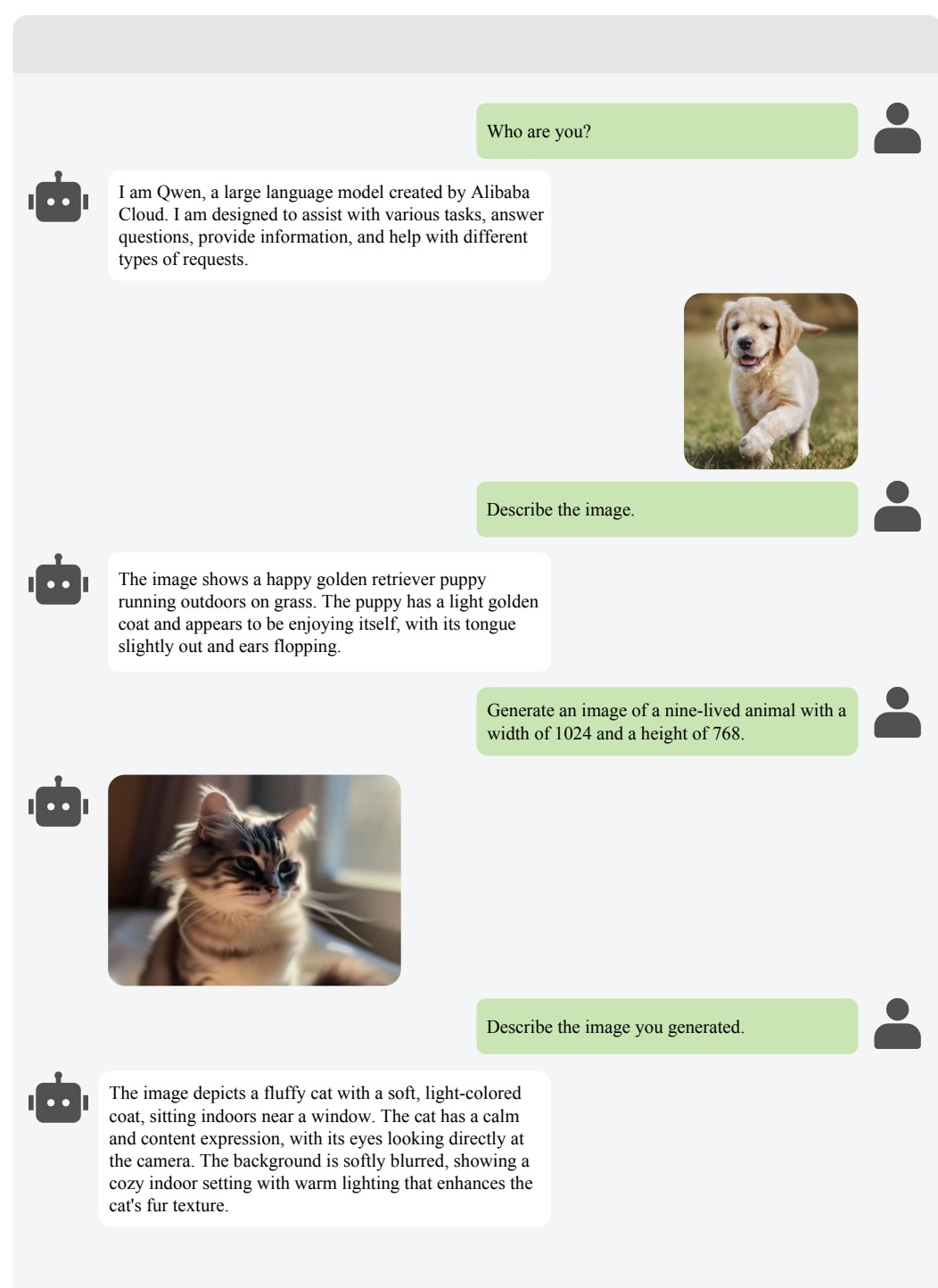

Figure 9: The interactive interface of the chat demo.

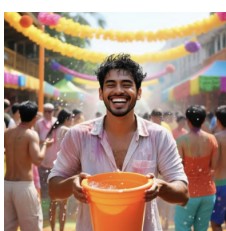 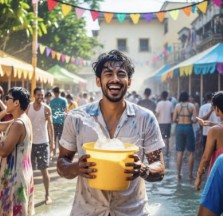 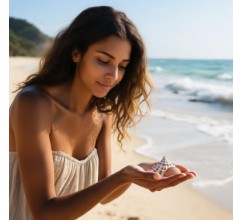 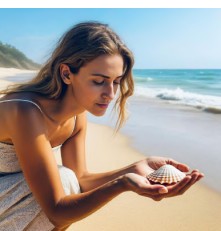

MeiGen-Image          MeiGen-Image-FaQ          MeiGen-Image          MeiGen-Image-FaQ

**Prompt:** A man is celebrating at a lively Water Splashing Festival, holding a basin and smiling joyfully, dressed in light summer clothing. The background features a square adorned with festive decorations, where colorful ornaments and cheerful crowds create a strong holiday atmosphere.

**Prompt:** A woman is coll-ecting seashells on a sunny beach, holding a beautiful shell in her hand. She is wearing a light beach dress, with a vast coastline in the background.

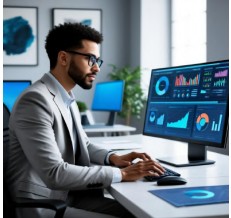 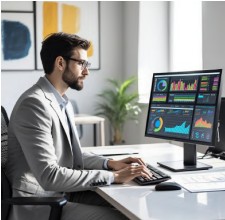 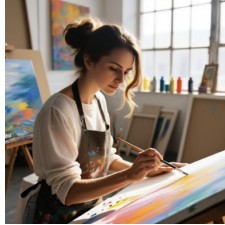 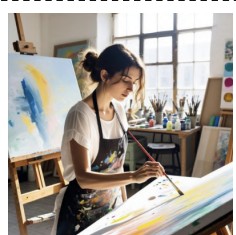

MeiGen-Image          MeiGen-Image-FaQ          MeiGen-Image          MeiGen-Image-FaQ

**Prompt:** An office worker in a light gray suit, wearing glasses, is typing rapidly on the keyboard. Various data charts are displayed on the computer screen.

**Prompt:** A woman is painting intently in a bright studio, holding a paintbrush and wearing an artist's apron, surrounded by paints and canvases. Sunlight streams through large windows, illuminating the entire space and creating an atmosphere filled with creativity and inspiration.

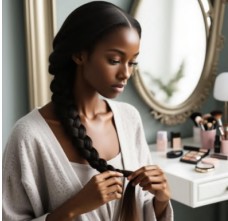 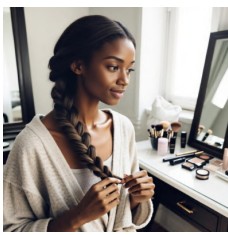 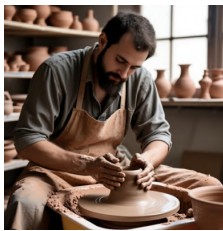 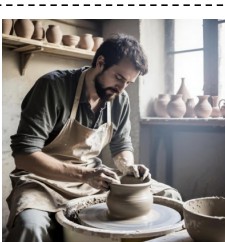

MeiGen-Image          MeiGen-Image-FaQ          MeiGen-Image          MeiGen-Image-FaQ

**Prompt:** A woman stands in front of a mirror, intently braiding her long hair, her fingers skillfully weaving through the strands. She is dressed in comfortable homewear, creating a relaxed and cozy atmosphere.

**Prompt:** A man is deeply focused on creating pottery in a ceramics studio, his hands skillfully shaping the clay and reflecting his passion and dedication to art.

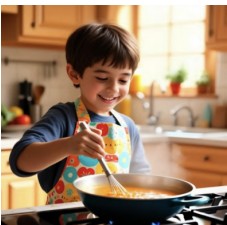 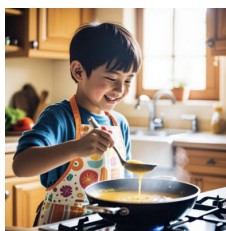 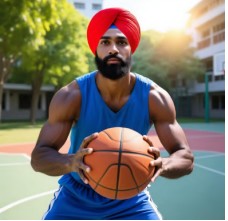 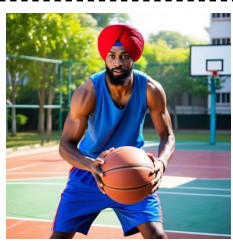

MeiGen-Image          MeiGen-Image-FaQ          MeiGen-Image          MeiGen-Image-FaQ

**Prompt:** A boy is cooking soup in a home kitchen, holding a stirring spoon with a smile on his face and wearing a brightly colored children's apron. The background features a warm-toned kitchen, with soft and cozy hues on the walls and cabinets.

**Prompt:** A muscular man wearing a red headband and a blue sports outfit holds a basketball, standing on a schoolyard court ready to make a shot.

Figure 10: Qualitative cases of MeiGen-Image and MeiGen-Image-FaQ (P1).

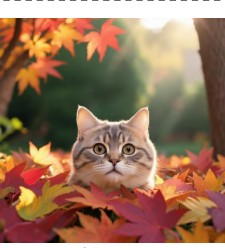 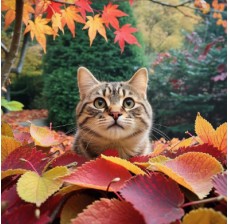 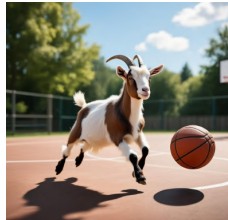 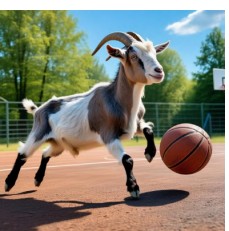

MeiGen-Image                MeiGen-Image-FaQ                    MeiGen-Image                MeiGen-Image-FaQ

**Prompt:** A curious cat peeking out from a pile of autumn leaves.

**Prompt:** A goat is playing basketball on a sunny out-door court. Its fur is fine and glossy, and its move-ments are agile, as if it is engaged in an intense game.

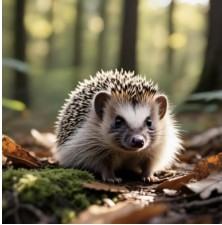 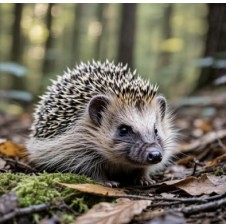 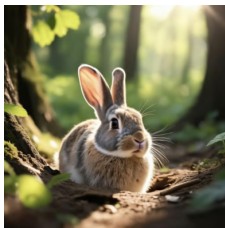 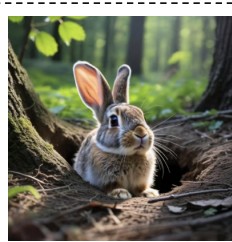

MeiGen-Image                MeiGen-Image-FaQ                    MeiGen-Image                MeiGen-Image-FaQ

**Prompt:** A hedgehog with quills displaying delicate textures and a natural sheen. The background is a tranquil forest floor, where fallen leaves and moss add a sense of natural texture to the scene.

**Prompt:** A rabbit has just poked its head out of a burrow, sunlight shining on its soft fur and giving it a warm glow. The background is a peaceful forest, lush and green, with dappled light filtering through the leaves.

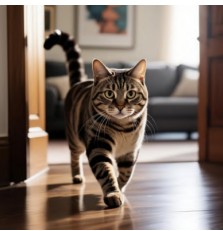 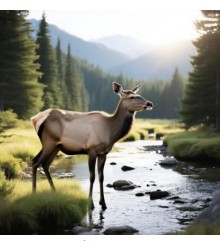 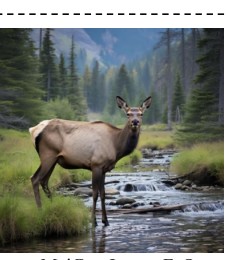

MeiGen-Image                MeiGen-Image-FaQ                    MeiGen-Image                MeiGen-Image-FaQ

**Prompt:** A tabby cat gracefully walks in through the doorway, its striped fur appearing especially distinct under the soft indoor lighting. The background features a cozy home setting, with several art paintings hanging on the walls and a dark wooden textured floor.

**Prompt:** An elk stands beside a mountain stream, surrounded by lush forest scenery. The stream is crystal clear, with sunlight sparkling on its surface.

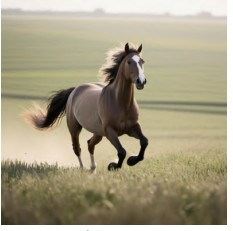 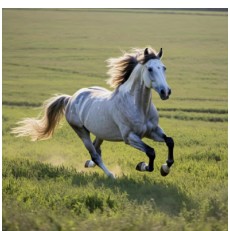 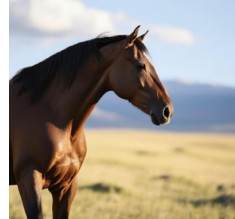 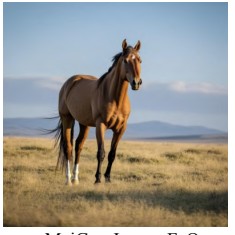

MeiGen-Image                MeiGen-Image-FaQ                    MeiGen-Image                MeiGen-Image-FaQ

**Prompt:** A horse is running across a vast field, captured from a bird's-eye view that highlights its agile form and the expansiveness of the landscape.

**Prompt:** A riderless horse stands on a vast grassland, sunlight shining on its smooth coat and reflecting a warm sheen.

Figure 11: Qualitative cases of MeiGen-Image and MeiGen-Image-FaQ (P2).

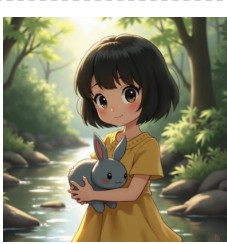 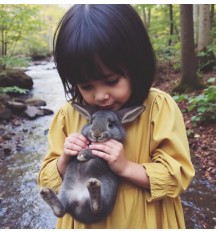 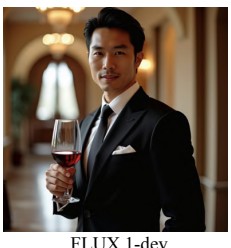 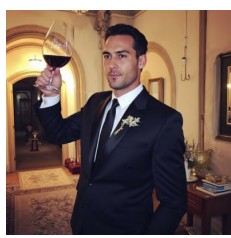

FLUX.1-dev          FLUX.1-dev-FaQ          FLUX.1-dev          FLUX.1-dev-FaQ

**Prompt:** A young girl, with short black hair, dressed in a yellow dress, holds a small gray rabbit with both hands, and the background is a forest stream with sunlight filtering through the leaves.

**Prompt:** A dashing man in a suit holding a glass of red wine with the backdrop of a luxurious mansion.

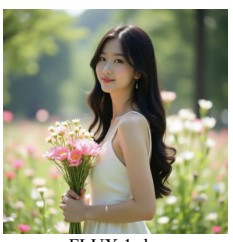 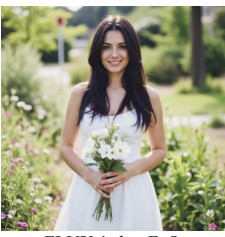 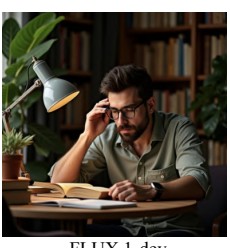 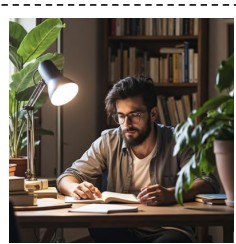

FLUX.1-dev          FLUX.1-dev-FaQ          FLUX.1-dev          FLUX.1-dev-FaQ

**Prompt:** A young woman with long black hair, holding a bouquet of fresh flowers, dressed in a sum-mer white dress, stands in a park brimming with bloom-ing flowers, with the sun-light casting its glow upon her face.

**Prompt:** A man reading in a cozy study, his right hand gently resting on his glasses, with warm light, wearing a casual shirt, against a backdrop filled with books and green plants.

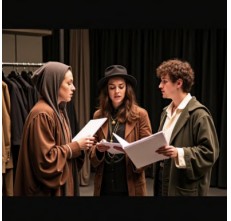 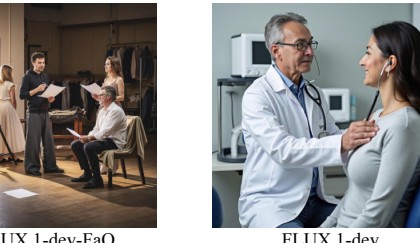 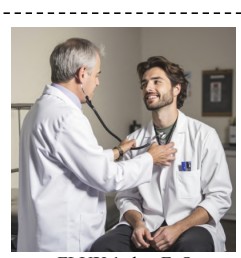

FLUX.1-dev          FLUX.1-dev-FaQ          FLUX.1-dev          FLUX.1-dev-FaQ

**Prompt** Actors whisper lines in the theater backstage, holding scripts and concentrating on practice, dressed in character costumes, with a backdrop of a theater costume rack.

**Prompt:** The physician is conducting a pulmonary examination on a patient in a welllit clinic, holding a stethoscope attentively against the chest, wearing a clean white lab coat, with a background of neat medical equipment.

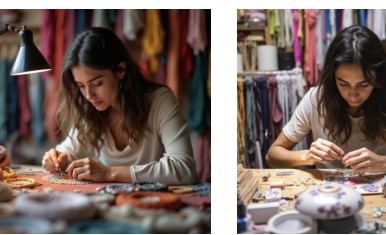 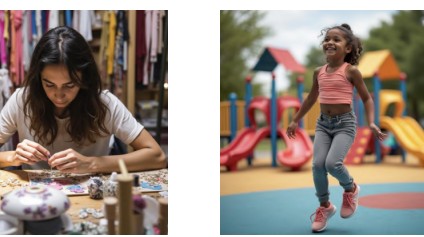 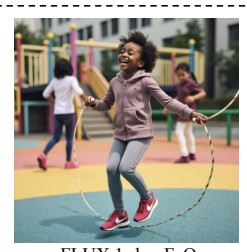

FLUX.1-dev          FLUX.1-dev-FaQ          FLUX.1-dev          FLUX.1-dev-FaQ

**Prompt:** A photograph capturing a lady immersed in the creation of unique art pieces at a hand-craft workshop. She is sewing with needles in hand, wearing a comfortable top that complements the creative atmosphere.

**Prompt:** A vibrant photograph capturing a girl joyfully jumping rope on a playground. She is dressed in sporty attire, her hands lightly and skillfully manipulating the rope with precision.

Figure 12: Qualitative cases of FLUX.1-dev and FLUX.1-dev-FaQ (P1).

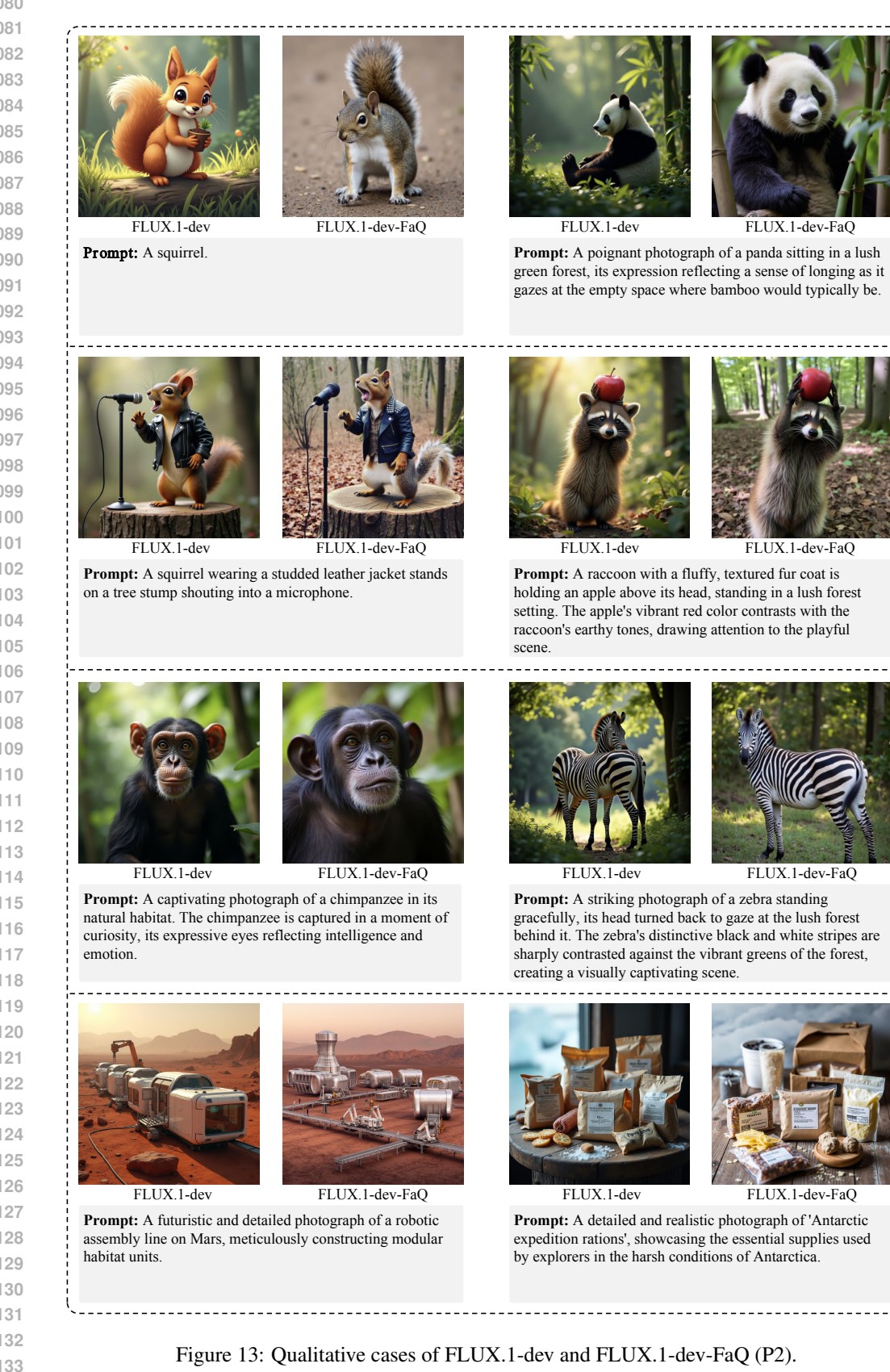

Figure 13: Qualitative cases of FLUX.1-dev and FLUX.1-dev-FaQ (P2).

