# OpenReview forum: "Forge-and-Quench: Enhancing Image Generation for Higher Fidelity in Unified Multimodal Models"
_ICLR.cc/2026/Conference — Submitted to ICLR 2026_

### Official Review · Reviewer_NTLm · 2025-10-21

**Soundness:** 2
**Presentation:** 3
**Contribution:** 2
**Rating:** 2
**Confidence:** 5

**Summary:**

The paper introduces Forge-and-Quench, a framework designed to enhance the fidelity and detail richness of images generated by Multimodal Large Language Models (MLLMs) and Text-to-Image (T2I) backbones. In the generation process, an MLLM first reasons over the entire conversational context, including text instructions, to produce an enhanced text instruction. This refined instruction is then mapped to a virtual visual representation, termed the Bridge Feature, via a Bridge Adapter. This feature acts as a link, refining the generation process.

**Strengths:**

1. The article is well-written, and the presentation is very clear.
2. The authors conducted experiments on multiple datasets, including not only automatic evaluation but also extensive qualitative analysis.

**Weaknesses:**

1. The novelty appears incremental: prompt recaptioning has already proven highly effective in works like DALL-E3 and Emu3, becoming common knowledge. Bridge Feature is essentially a SigLIP representation, and the adapter resembles a tiny diffusion model. Prior efforts (e.g., BLIP3-o) have denoised CLIP features to condition another diffusion model for image generation. The paper reads more like an engineering refinement of BLIP3-o via recaptioning tricks, wrapped in a “Forge-and-Quench” narrative.
2. The main experiments pit the method against a pure diffusion baseline. Today’s leading image-generation models (e.g., Qwen-Image) already leveraged the visual-understanding power of MLLMs to improve generation. Simply introducing an MLLM already boosts DiT performance, so I remain skeptical of the gains attributed to the authors’ intricate designs; more competitive baselines should be included.
3. Performance on Geneval actually drops. Since Geneval prompts are short, prompt rewriting should help (as shown in Emu3). A decline seems to suggest that the enhanced prompt and Bridge Feature fed into the DiT do not act synergistically.
4. If we only recaption prompts, how well does a training-free approach—feeding the rewritten text directly into a freeze DiT such as flux—perform on Geneval and WISE? The authors should analyze this. For example, WISE already provides rewritten prompts; I think authors can compare t2i performance directly using these prompts for freezed DiT against the original prompt + proposed FaQ framework.
5. With the main baseline already limited, ablation studies are crucial. I do not see why COCO-30K FID was chosen as the sole metric for these ablations.

**Questions:**

Please refer to weaknesses.

---

> ### Author Response · Authors · 2025-12-02
> **Response to Weaknesses 1**
>
> One of the core contributions of this work is the discovery that features obtained via the Bridge Adapter can significantly enhance the visual quality of generated images (realism and detail representation), which has not been explored in prior work.  For generating SigLIP features, our approach is similar to Blip-3o: both regress SigLIP features from the prompt using diffusion. However, we validate through ablation studies that diffusion is the optimal architecture for this task, and Blip-3o does not show this. Furthermore, while Blip-3o directly uses the generated SigLIP features as a condition for the SANA model to generate images, we inject the SigLIP features into a standard T2I diffusion model in a manner similar to IP-Adapter. This approach enables us to observe significant improvements in image realism and detail representation.

---

> ### Author Response · Authors · 2025-12-02
> **Response to Weaknesses 2**
>
> This work highlights the "improvement in the visual quality of generated images", this enhancement is strongly correlated with the choice of adapter features (SigLIP-ViT features contribute to noticeable improvements, whereas Qwen-VL-ViT features do not yield significant gains). We introduce the MLLM primarily to better regress the SigLIP features; however, it is still possible to regress adapter features without using an MLLM, although the performance is somewhat reduced.

---

> ### Author Response · Authors · 2025-12-02
> **Response to Weaknesses 3**
>
> For the GenEval benchmark, to ensure fairness, we use the original prompts (non-enhanced prompts) for inference across all models. This information was not mentioned in the paper but will be added in a future update.

---

> ### Author Response · Authors · 2025-12-02
> **Response to Weaknesses 4**
>
> Using the enhanced text alone improves the model's world knowledge reasoning capability, as reflected by performance gains on the WISE benchmark, but does not result in noticeable improvements in the visual quality of generated images. Conversely, using only the Bridge Feature significantly enhances the visual quality of generated images, but does not improve world knowledge reasoning capability.

---

> ### Author Response · Authors · 2025-12-02
> **Response to Weaknesses 5**
>
> The FiD metric shows indeed bias or deviation in text-to-image generation tasks, and human annotation is a more reliable indicator. However, due to cost constraints, we only conducted human side-by-side (SBS) evaluations in the main experiments, while relying solely on FiD for ablation studies.

---

### Official Review · Reviewer_ifvy · 2025-10-22

**Soundness:** 3
**Presentation:** 3
**Contribution:** 3
**Rating:** 6
**Confidence:** 4

**Summary:**

This paper proposes a method that leverages a Bridge Feature to enrich semantic information in the text-to-image generation process. Without relying on any additional reference images, the approach produces high-quality visual features to guide the image synthesis, thereby improving both the fine-grained detail and overall quality of the generated images. The authors validate their method on two T2I backbones — FLUX.1-dev and MeiGen-Image — and evaluate its performance across five benchmarks, measuring improvements in terms of image quality, instruction-following capability, and other relevant dimensions.

**Strengths:**

* The paper introduces a novel Bridge Feature approach that generates high-quality visual features from text alone, effectively bridging the gap in scenarios where traditional T2I methods rely on real image features.
* The proposed framework is modular, with both the MLLM and T2I backbones kept frozen while only lightweight adapters are trained, which reduces training cost and improves portability across different model architectures.
* The method is validated on two distinct T2I backbones and evaluated using five benchmarks covering various dimensions.

**Weaknesses:**

* The composition of the training data is not explicitly discussed, which makes it difficult to assess the proposed method’s performance in both in-domain and out-of-domain scenarios. Since the Bridge Feature requires supervised training, further comparison between the proposed method and the base model in out-of-domain cases would be necessary to understand its generalization capability.

* The paper mentions that input prompts are first expanded into Enhanced Text $ t^* $, a process which empirically may alter the fine-grained details of the generated images. However, the potential impact of this semantic expansion on image fidelity and alignment with user intent is not sufficiently discussed in the experimental section.

**Questions:**

* Could the authors clarify the domain composition of the training dataset? Are there any experimental results that explicitly evaluate the proposed method’s ability to generalize across domains?

* In the Forge-and-Quench framework, what is the performance when only the Enhanced Text $ t^* $ is used as input, or when only the Bridge Feature is used, without the other? Such an ablation would help quantify the individual contributions of semantic enrichment and synthetic visual features to the final image quality.

* Since the Injection Adapter needs to be trained specifically for a given T2I backbone, this design could limit its generalization to other architectures. Could the authors provide more details on the adapter’s training cost, and how it compares to the overall framework’s training complexity?

---

> ### Author Response · Authors · 2025-12-02
> **Response to Question 1**
>
> We trained the Bridge Adapter module on 200 million image-text pairs, with the training data primarily sourced from open datasets such as LAION-5B and DataComp, as well as some proprietary data. Our proposed method demonstrates improved visual quality of generated images across different T2I diffusion models, indicating a certain level of generalization capability.

---

> ### Author Response · Authors · 2025-12-02
> **Response to Question 2**
>
> Using the enhanced text alone improves the model's world knowledge reasoning capability, as reflected by performance gains on the WISE benchmark, but does not result in noticeable improvements in the visual quality of generated images. Conversely, using only the Bridge Feature significantly enhances the visual quality of generated images, but does not improve world knowledge reasoning capability.

---

> ### Author Response · Authors · 2025-12-02
> **Response to Question 3**
>
> In the Forge-and-Quench framework, the Forge module (which generates adapter features from text) only needs to be trained once, requiring 48 hours on 32 H100 GPUs. In contrast, the Quench module (which injects adapter features into the T2I diffusion model) must be adapted for each base model, requiring 24 hours of training on 32 H100 GPUs per model.

---

### Official Review · Reviewer_x7Vi · 2025-10-30

**Soundness:** 3
**Presentation:** 3
**Contribution:** 2
**Rating:** 2
**Confidence:** 5

**Summary:**

This paper introduces Forge-and-Quench, a unified multimodal framework that aims to enhance the fidelity and detail quality of generated images by tightly coupling an MLLM with a Text-to-Image diffusion backbone.
The central insight is that current unified multimodal models (e.g., MetaQuery, BLIP3-o) treat the MLLM as a static prompt rewriter, passing only high-level semantic embeddings to the diffusion model. The authors argue this creates an informational bottleneck that limits fine-grained detail control. Extensive experiments on FLUX.1-dev and MeiGen-Image backbones show that Forge-and-Quench substantially improves FID, GPT-Fidelity, and visual realism, while maintaining comparable prompt-alignment (GenEval, DPG-Bench) and even enhancing world-knowledge reasoning (WISE).

**Strengths:**

1.The paper’s modular adapter design elegantly separates understanding (Forge) and generation (Quench), enabling compatibility with a variety of existing MLLM and T2I models without requiring joint pre-training or large-scale end-to-end optimization.
2.The method demonstrates consistent qualitative improvements in visual fidelity and texture realism across both tested backbones, showing that injecting a virtual visual feature can strengthen fine-grained detail synthesis while maintaining reasonable instruction alignment.

**Weaknesses:**

1.The enhanced-text stage is not novel, as similar reasoning-based prompt enrichment has been extensively studied in earlier works such as Bagel, T2I-R1 and GoT, yet the paper does not acknowledge or discuss these predecessors specifically. The bridge-feature component largely replicates what Emu2, Seed-X, PUMA have already done with CLIP-space feature injection, and the paper fails to analyze how its approach differs conceptually.
2. The method performs worse on GenEval and DPG-Bench benchmarks, but the paper does not provide any analysis or explanation for these drops.
3.The qualitative section does not actually show the MLLM’s internal reasoning or thinking process, leaving the “understanding-to-generation” connection mostly unsubstantiated.
4.The approach is highly sensitive to the choice of target visual feature space, with ablations showing large disparities and pronounced noise fragility for SigLIP2, yet the paper provides no principled rationale or robustness strategy beyond defaulting to SigLIP-ViT.
5.The qualitative evaluation includes only one example per prompt, which makes the comparison potentially cherry-picked rather than representative. When I attempted to reproduce examples with FLUX, the “distorted zebra” artifact highlighted in the paper turned out to be rare and atypical.

**Questions:**

See weaknesses

---

> ### Author Response · Authors · 2025-12-02
> **Response to Weaknesses 1**
>
> One of the core contributions of this work is the discovery that understanding features obtained via the Bridge Adapter can significantly enhance the visual quality of generated images (realism and detail representation), which has not been explored in prior work. In contrast, existing methods such as Bagel focus on enhancing text (e.g., incorporating world knowledge) to improve text-image alignment, but do not address the visual quality of generated images.

---

> ### Author Response · Authors · 2025-12-02
> **Response to Weaknesses 2**
>
> As shown in Table 1, our model demonstrates a significant advantage over the base model in visual quality (COCO-30K FID, GPT-Fidelity), while achieving comparable performance to the base model in text-image alignment (GenEval, DPG-Bench), which is consistent with our expectations. These results indicate that the proposed Bridge Adapter significantly improves the visual quality of generated images without compromising the text-image alignment capability of the base model.

---

> ### Author Response · Authors · 2025-12-02
> **Response to Weaknesses 3**
>
> We found that the features obtained via the Bridge Adapter can significantly enhance the visual quality of generated images, and the acquisition of these features may not involve the MLLM’s internal reasoning or thinking process.

---

> ### Author Response · Authors · 2025-12-02
> **Response to Weaknesses 4**
>
> We conducted ablation studies on several ViT features and found that SigLIP-ViT achieves the best results in improving visual quality and reducing distortion. These effects are highly correlated with the specific ViT features employed. The robustness of SigLIP2 features is relatively poor, which we suspect is related to its objective function—SigLIP2 introduces an additional NTP loss on top of SigLIP, potentially resulting in larger feature magnitudes and decreased robustness. We plan to explore this in the future work.

---

> ### Author Response · Authors · 2025-12-02
> **Response to Weaknesses 5**
>
> We found that the features obtained via the Bridge Adapter can enhance the visual quality of generated images. This conclusion is robust and generalizable. Additional cases illustrating this improvement are provided in Appendix 4.

---

### Official Review · Reviewer_r2dT · 2025-10-30

**Soundness:** 3
**Presentation:** 3
**Contribution:** 2
**Rating:** 6
**Confidence:** 4

**Summary:**

The paper introduces Forge-and-Quench, a framework that uses an MLLM to refine instructions and a Bridge Adapter to convert them into a latent visual “Bridge Feature,” guiding a T2I model toward higher fidelity and detail. The authors claim that this structure effectively transfers understanding-derived visual insights into the generation pipeline. This design is also compatible with different MLLMs and T2I backbones while minimizing training costs and preserving the original model’s multimodal understanding ability. Experiments across two models show improved image fidelity, richer details, and better instruction-following behavior, along with enhanced use of world knowledge.

**Strengths:**

1. The paper is clearly written, and the overall framework is easy to understand and follow.

2. The observation that introducing a reference image can improve fidelity and detail richness in generated images is insightful and valuable. The experimental results effectively demonstrate this contribution.

**Weaknesses:**

1. The primary weakness lies in the design of how the reference image is created. The authors propose using a Bridge Adapter to map the MLLM-generated text into a SigLIP image feature via a rectified flow model, but this design seems unintuitive. The mapping step still introduces information loss. In contrast, BLIP-3o directly generates the SigLIP image embedding in a rectified flow manner via their unified model and then generate a high-fidelity image from that embedding, which is more consistent and avoids this loss. In fact, BLIP-3o’s architecture naturally follows the same conceptual pipeline as the authors’ idea, first produce an abstract image representation, then generate it.

2. Another concern is that the comparison is insufficient. The authors only compare against their base model and omit comparisons with alternative frameworks that achieve similar functionality, such as BLIP-3o. Including such comparisons or at least providing discussion would strengthen the argument for effectiveness.

3. A minor weakness: the authors should clarify why their method achieves better performance on the WISE benchmark, which focuses on reasoning-driven image generation. The current results suggest that the improvement may come primarily from the MLLM component rather than from the proposed architectural innovations.

At this stage, I lean toward a weak accept because the paper provides interesting insights. If the authors can satisfactorily address the issues raised in my review, I would be open to maintaining or even improving my score.

**Questions:**

1. Is the proposed reference-image creation pipeline better than BLIP-3o?
Since the Bridge Adapter still introduces information loss when mapping text → SigLIP feature, why not directly generate SigLIP features as BLIP-3o does? What concrete advantages does the proposed design offer?

2. Why does the method outperform on the WISE benchmark?
Is the performance gain mainly due to the MLLM component, or is it attributable to the proposed Bridge Adapter  design? Please clarify the source of improvement.

3. Where is the citation of the backbone, MeiGen-Image and FLUX.1-dev?
The paper does not clearly cite the backbone models used, MeiGen-Image and FLUX.1-dev, especially MeiGen-Image which is difficult to locate or search online. Please provide proper citations or references (e.g., paper / project page / GitHub link) and clarify how these backbones are used in your framework.

---

> ### Author Response · Authors · 2025-12-02
> **Response to Question 1**
>
> Our method for generating SigLIP features is similar to Blip-3o. Both methods use diffusion to predict SigLIP features from the prompt. But unlike Blip-3o, we prove through ablation studies that diffusion is the best choice for this task. Blip-3o does not show this. Also, Blip-3o uses the generated SigLIP features directly as a condition for the Sana model. We do something different. We add the SigLIP features to a standard T2I diffusion model, similar to how IP-Adapter works. This helps us get better realism and more details in the generated images.

---

> ### Author Response · Authors · 2025-12-02
> **Response to Question 2**
>
> The gain on the WISE benchmark comes from using MLLM for PE. It is not due to the Bridge Adapter we propose in this work.

---

> ### Author Response · Authors · 2025-12-02
> **Response to Question 3**
>
> For FLUX.1-dev, we give a proper citation in Chapter 2 (Related Works). For MeiGen-Image, this is a 6B-parameter model developed internally. We plan to release it soon and will add the citation once it is available.

---

### Meta-Review · Area_Chair_A6M5 · 2026-01-04

**Summary:**

The submission presents Forge-and-Quench, a framework that leverages MLLM for prompt enhancement and a Bridge Adapter to generate virtual visual features, aiming to boost image fidelity in multimodal generation. While the authors responded, they failed to adequately address core weaknesses raised by reviewers. Key concerns include insufficient novelty relative to prior work (e.g., BLIP-3o, Emu2), unsubstantiated claims about the framework’s unique contributions, inadequate analysis of performance tradeoffs (e.g., GenEval score drops), and lack of rigorous validation for generalization and component synergies. The authors’ rebuttals largely avoid direct engagement with critical issues, leaving fundamental questions unresolved.

**Reviewer Concerns:**

- **Addressed Concerns**: Minimal meaningful addressing of core weaknesses. The authors clarified training data sources, adapter training costs, and partial details about backbone citations.

- **Outstanding Concerns**:
1. Novelty gaps: The framework’s core components closely mirror prior work, with no compelling demonstration of unique methodological advances.
2. Insufficient comparisons: Lack of evaluation with relevant baselines (e.g., BLIP-3o, Qwen-Image) undermines claims of superiority.
3. Unanalyzed tradeoffs: The paper provides no substantive explanation for drops in GenEval performance.
4. Weak ablation and validation: The authors did not address requests to quantify individual component contributions (e.g., enhanced text vs. Bridge Feature alone) or validate generalization to out-of-domain scenarios.

**Reviewer Scores:**

- Reviewer r2dT (Score: 6): Would likely **lower the score to 4** because the authors did not respond to the primary weakness.
- Reviewer x7Vi (Score: 2): Would retain the rejection score, as key concerns about prior work acknowledgment and result representativeness remain unaddressed.
- Reviewer ifvy (Score: 6): Would likely **lower the score to 4** given the lack of discussion on the proposed concerns.
- Reviewer NTLm (Score: 2): Would retain the rejection score, as the authors failed to address baseline inadequacies and ablation limitations.

---

### Decision · Program_Chairs · 2026-01-26

Reject